# ZmADF5, a Maize Actin-Depolymerizing Factor Conferring Enhanced Drought Tolerance in Maize

**DOI:** 10.3390/plants13050619

**Published:** 2024-02-24

**Authors:** Bojuan Liu, Nan Wang, Ruisi Yang, Xiaonan Wang, Ping Luo, Yong Chen, Fei Wang, Mingshun Li, Jianfeng Weng, Degui Zhang, Hongjun Yong, Jienan Han, Zhiqiang Zhou, Xuecai Zhang, Zhuanfang Hao, Xinhai Li

**Affiliations:** 1State Key Laboratory of Crop Gene Resources and Breeding, Institute of Crop Sciences, Chinese Academy of Agricultural Sciences, Beijing 100081, China; liubojuan1995@163.com (B.L.); yangruisixj@163.com (R.Y.); 15932271022@163.com (X.W.); luoping987@126.com (P.L.); chenyong9576@126.com (Y.C.); wangfei19990908@163.com (F.W.); limingshun@caas.cn (M.L.); wengjianfeng@caas.cn (J.W.); zhangdegui@caas.cn (D.Z.); yonghongjun@caas.cn (H.Y.); hanjienan@caas.cn (J.H.); zhouzhiqiang@caas.cn (Z.Z.); 2State Key Laboratory of North China Crop Improvement and Regulation, College of Agronomy, Hebei Agricultural University, Baoding 071000, China; wangnan@hebau.edu.cn; 3International Maize and Wheat Improvement Center (CIMMYT), Apdo. Postal 6-641, Texcoco 06600, Mexico; xc.zhang@cgiar.org

**Keywords:** maize (*Zea mays* L.), drought tolerance, *ZmADF5*, transcriptome analysis

## Abstract

Drought stress is seriously affecting the growth and production of crops, especially when agricultural irrigation still remains quantitatively restricted in some arid and semi-arid areas. The identification of drought-tolerant genes is important for improving the adaptability of maize under stress. Here, we found that a new member of the actin-depolymerizing factor (ADF) family; the *ZmADF5* gene was tightly linked with a consensus drought-tolerant quantitative trait locus, and the significantly associated signals were detected through genome wide association analysis. *ZmADF5* expression could be induced by osmotic stress and the application of exogenous abscisic acid. Its overexpression in *Arabidopsis* and maize helped plants to keep a higher survival rate after water-deficit stress, which reduced the stomatal aperture and the water-loss rate, as well as improved clearance of reactive oxygen species. Moreover, seventeen differentially expressed genes were identified as regulated by both drought stress and *ZmADF5*, four of which were involved in the ABA-dependent drought stress response. *ZmADF5*-overexpressing plants were also identified as sensitive to ABA during the seed germination and seedling stages. These results suggested that *ZmADF5* played an important role in the response to drought stress.

## 1. Introduction

Abiotic stress, especially drought, seriously affects crop growth, survival, and productivity. The steady growth of production is the primary goal for crop drought-tolerance breeding under the influence of climate change. As reduced variation in grain yield (GY) is a response to stress, the selection of superior genotypes for GY under drought-stressed conditions usually has limited and inconsistent progress [1]. Linkage-based quantitative trait loci (QTL) mapping with bi-parent populations enables the identification of chromosomal regions associated with these quantitative traits of tolerance and their further application to quick genome selection. So far, numerous QTL that control stress tolerance have been identified [2,3,4,5]. Meta-analysis helps to combine and compare these QTL identified from independent analyses to determine the number and location of existing, consensus QTL. Therefore, the confidence interval of an integrated QTL cluster can be reduced to improve the accuracy and effectiveness of QTL mapping with few confirmed QTL [2,6]. Linkage disequilibrium-based association analyses with natural populations is usually performed as a complement to linkage-based QTL mapping. So, joint linkage mapping and association analyses have been applied in the detection of QTL or genes for complex traits, including drought tolerance [7,8,9]. Up to now, a lot of QTL/genes of important traits have been identified using combined linkage mapping and association analyses in maize, including genes related to plant height and ear height [10], male inflorescence size [11], gray leaf spot disease resistance [12], waterlogging tolerance [13], and drought tolerance [7]. However, few of them have been cloned or functionally analyzed.

The actin-depolymerizing factor (ADF) gene family is an actin-binding protein family and a major regulator of actin dynamics in eukaryotes [14,15]. ADFs can incorporate globular actin (G-actin) into filamentous actin (F-actin) to extend and maintain the stability of actin by nucleation; alternatively, they also can sever F-actin by dissociation [16]. Both processes are ATP-dependent and affected by several factors such as pH [17] and phosphorylation of the ADFs [18]. More ADFs have been identified in plants than in animals recently, accompanied by more diverse biochemical functions caused by mutations during evolution [15,16,19]. Currently, the ADF gene family has been reported to play a crucial role in plant growth and development, including hypocotyl and root hair elongation, pollen germination and pollen tube growth, flowering time, cotton fibre development, etc. [20,21,22,23,24]. For example, down regulation of *AtADF1* gene expression affects flowering time [23]. The *AtADF5* loss-of-function mutants exhibited delayed pollen germination and defected pollen tubes [21]. Moreover, several ADFs in plants have also been reported in relation to the regulation of biotic or abiotic stress responses. In *Arabidopsis*, *AtADF2* is involved in resistance to root-knot nematode infection [25]. *AtADF4* links pathogen perception and defense activation [26], as well as drought stress tolerance [27]. *AtADF5* is another drought stress-responsive gene that responds to drought stress by regulating stomatal closure [28]. The overexpression of *OsADF3* in rice enhances drought tolerance in *Arabidopsis* [29]. In wheat, *TaADF3* [16], *TaADF4* [30], and *TaADF7* [31] play an important role in disease resistance. However, few ADF genes have been identified in maize, especially in association with biotic or abiotic stress resistance.

In this study, we identified and characterized a maize ADF gene, named *ZmADF5*, originally derived from a consensus drought-tolerance QTL in maize. To clarify whether and how *ZmADF5* participates in drought stress, we analyzed the phylogenetic relationship of ADF in different species and the expression pattern of the *ZmADF5* gene under drought stress. Meanwhile, we generated overexpressed and knockout *ZmADF5* transgenic plants. We found that *ZmADF5*-overexpressing transgenic plants displayed enhanced tolerance to drought. Transcriptome data analysis and susceptibility to ABA in *Arabidopsis thaliana* with overexpression of *ZmADF5* suggest that *ZmADF5* also enhances drought tolerance through an ABA-dependent pathway. Our research shows that *ZmADF5* could be a crucial candidate gene for improving drought tolerance in maize.

## 2. Results

### 2.1. Identification of the Relationship between ZmADF5 and Two Linked Polymorphic Markers

Drought-tolerant consensus QTL 1 (DCQ1) was previously identified as located on maize chromosome bin 1.03 (Figure 1A, adapted from Hao et al., 2010 [2]). Association analysis with 538 known markers was used for the mapping of DCQ1 to identify the drought-tolerance genes near bin 1.03. In total, 517 protein-coding genes were identified with a mean of 1.04 markers per gene. Ten significant association signals were detected in relation to GY at the *p* = 10^−3^ level, but only one single nucleotide polymorphism marker (PZE-101047611) was identified with the GY trait during exposure to the water-stressed (WS) condition in Xinjiang in 2009 (Figure 1B); this marker explained 9.55% of the GY variation under WS and was located on the 5′-untranslated region of the *GRMZM2G077942* gene. Notably, the tightly linked polymorphic marker umc1073 was identified on the same gene by linkage mapping within the DCQ1 confidential interval peak. Therefore, the gene *GRMZM2G077942* might be a candidate gene for tolerance to drought stress in maize. *GRMZM2G077942* encodes a conserved actin-binding protein (ABP) (referred to hereafter as *ZmADF5*), belonging to an ubiquitous family of low molecular weight (15~20 kDa).

### 2.2. Phylogenetic Analysis of the ADF Gene Family in Maize and Other Plant Species

*ZmADF5* belongs to the ADF family, which includes the main regulators of actin in plants. The functional divergence of ADFs has been discussed in *Arabidopsis* [19]. To examine the phylogenetic relationships among ADFs in maize, *Arabidopsis*, and other plant species, an NJ tree was constructed after the alignment of 78 full-length ADF amino acid sequences in seven species (Appendix A). Except for *ZmADFn-2* and *GmADF1*, all ADFs were clustered into four groups, which supported a previous classification [15]. Groups I, II, and IV were composed of ADFs from eudicot and monocot plants clustered together. Moreover, Group III exclusively comprised monocots (Appendix A). Phylogenetic analysis showed that *ZmADF1*, *ZmADF2-2*, *ZmADFn-3*, *ZmADFn-4*, and *ZmADFn-5* clustered in Groups I; *ZmADF4* clustered in Groups II; *ZmADF3-1*, *ZmADF3-2*, and *ZmADF3-3* clustered in Groups III; and *ZmADF2-1*, *ZmADF5-1*, *ZmADF5-2*, *ZmADF6*, and *ZmADFn-1* clustered in Groups IV. A distinct difference in the length of the first exon of the ADFs was detected among the four groups. The first exon of the ADFs clustered in Groups I, II, and III contained only three bases (ATG) (Appendix A), which was the initiation codon; the length of the first exon varied in the Group IV ADFs [19]. To further characterize the diversity of the ADF gene family, we analyzed the distributions of conserved motifs. Twenty motifs were identified using the MEME website (Appendix A). Motifs 1–4 were present in most of the ADF proteins, while the other conserved motifs were group specific. For example, motifs 5–7 were only found in Group IV.

Among the ADFs in maize, two were annotated with *ZmADF5* at the NCBI: one was located on chromosome 1 (*ZmADF5-1*) and the other was located on chromosome 9 (*ZmADF5-2*). These two genes had an almost identical amino acid sequence with a similarity of 80.81%, except for the additional 29 amino acids in the N-terminal of *ZmADF5-2* (Appendix A). Both of the *ZmADF5* genes were clustered in Group IV, where almost all ADF5 and ADF9 sequences were included (Appendix A). *AtADF5* and *AtADF9* have opposing biochemical properties compared to other ADFs in *Arabidopsis* because of N-terminal extensions and amino acid changes during evolution [19]. Moreover, *AtADF5* [32] and *AtADF9* [33] control development in *Arabidopsis*; *AtADF5* is involved in stress response to cold [32] and drought [28]. These results indicate that *ZmADF5* is involved in plant development and/or the stress response.

### 2.3. ZmADF5 Expression Is Induced by Drought Stress

*ZmADF5* was expressed in almost all tissues in maize (e.g., roots, stems, leaves, silk, and tassels) throughout the growth period; however, it had higher expression levels in leaf tips (Figure 2A). Under the water-deficit treatment, the gene was up-regulated in root and leaf tissues; the corresponding maxima were 3.60-fold and 5.36-fold greater than in the control (0 h), respectively (Figure 2B). Under the ABA treatment, the gene was up-regulated in root, stem, and leaf tissues at all time points; the corresponding peaks were present at 1, 3, and 1 h, respectively (Figure 2C). These results indicate that *ZmADF5* might be induced in the drought- and ABA-stress responses.

### 2.4. Subcellular Localization of ZmADF5

To determine the subcellular localization of ZmADF5, a 35S::*ZmADF5*-*GFP* fusion construct was generated and transformed into maize protoplasts. Confocal scanning analysis showed that the fluorescence of 35S::*GFP* was distributed throughout the cell, while the fluorescence of 35S::*ZmADF5-GFP* was present in the nucleus and the cytosol in maize protoplasm (Figure 2D).

### 2.5. Drought Tolerance Is Enhanced in Arabidopsis Transgenic Plants through Overexpression of ZmADF5

Three independent *ZmADF5*-overexpressing transgenic lines of *Arabidopsis* were used for further analysis. After the cessation of watering for 3 weeks, most leaves in the wild type (WT) and transgenic plants began to dry and become dark because of water loss. Most WT plants gradually withered during the water-deficit stress, while the transgenic plants remained green and few withered (Figure 3A). Survival rate analysis in *Arabidopsis* exposed to water-deficit stress revealed that 82% of the transgenic plants were alive, whereas all WT plants had died (Figure 3B). As reported, *Arabidopsis AtADF5* promotes stomatal closure in response to drought stress [28]. Water-loss assays were conducted on detached leaves to test whether *ZmADF5* is also involved in the regulation of stomatal movements under drought stress. As shown in Figure 3C, water loss was slower in *ZmADF5*-overexpressing leaves than in WT leaves under dehydrating conditions. No significant differences in the stomatal aperture were detected between the WT and transgenic plants under normal-watering conditions. However, the stomatal aperture after water-deficit treatment was larger in WT plants than in transgenic plants (Figure 3D,E). These results indicate that overexpression of *ZmADF5* might improve drought tolerance in *Arabidopsis*.

### 2.6. Overexpression of ZmADF5 Enhances Drought Tolerance in Maize

To confirm that *ZmADF5* enhanced drought tolerance, we constructed *ZmADF5*-overexpressing (*ZmADF5*-OE) lines and *ZmADF5*-knockout (*ZmADF5*-KO) lines of maize, which were subjected to drought-tolerance assays (Appendix A and Appendix A). The *ZmADF5* transcript levels in the two controls (Zheng58 and C01), as well as the *ZmADF5*-OE and *ZmADF5*-KO lines, are presented in Figure 4B and Figure 5B. After 10 days of water-deficit stress, most control plants (Zheng58) began to wither, while only a few leaves rolled and wilted in the *ZmADF5*-OE lines. The *ZmADF5*-OE lines appeared normal with leaves that remained green and grew better than the control plants did (Figure 4A); however, the *ZmADF5*-KO plants showed the opposite result (Figure 5A). The survival rates of the two controls and the transgenic lines were recorded 3 days after rehydration. The survival rates of the OE lines were 2.30–2.50-fold greater than the survival rate of Zheng58 (Figure 4C), while the survival rates of the KO lines were 1.50–1.97-fold lower than that of C01 (Figure 5C). The water-loss rate of the OE lines was significantly lower than the water-loss rates of the Zheng58 plants at the beginning 1–4 h; the greatest difference happened at 4 h (Figure 4D), while the *ZmADF5*-KO plants showed opposite results (Figure 5D).

Furthermore, we surveyed some physiological parameters (including malondialdehyde (MDA) and anthocyanin contents, as well as catalase (CAT) and superoxide dismutase (SOD) activities, chlorophyll content) of the two controls and the transgenic plants under normal-watering and water-deficit treatments at the same time. No significant differences in five physiological parameters were observed between the control and transgenic plants in the norma-watering treatment; under the water-deficit treatment, the activities of SOD and CAT, as well as the chlorophyll content, were significantly greater in *ZmADF5*-OE lines than in control Zheng58 plants (Figure 4E–G). These physiological parameters were significantly lower in *ZmADF5*-KO plants than in C01 plants (Figure 5E–G). Compared with the normal-watering treatment, anthocyanin and MDA contents significantly accumulated in the transgenic and control plants under continuous water-deficit treatment, but *ZmADF5*-OE plants exhibited less anthocyanin and MDA accumulation than Zheng58 (Figure 4H,I) did; *ZmADF5*-KO plants exhibited more anthocyanin and MDA accumulation than C01 plants did (Figure 5H,I). This indicated that the drought tolerance of *ZmADF5* might have involved the clearance of reactive oxygen species (ROS) after suffering drought damage.

### 2.7. ABA Dependence of Enhanced Drought Tolerance in Transgenic Plants

RNA-Seq was performed to further explore the mechanism of the improved drought tolerance mediated by *ZmADF5* overexpression. We compared gene expression patterns in the *ZmADF5*-overexpressing (OE10) transgenic line and WT plants of *Arabidopsis* under normal-watering and water-deficit treatments (WT-N and OE-N under normal-watering conditions; WT-D and OE-D under water-deficit conditions). In total, 319 differentially expressed genes (DEGs) were identified in the normal-watering condition, with 252 up-regulated and 67 down-regulated in the OE10 line relative to the WT (Figure 6A). Under water-deficit stress conditions, 55 genes were differentially expressed (34 up-regulated and 21 down-regulated) (Figure 6B). Subsequently, Gene Ontology (GO) analysis was performed to identify the potential functions of these genes. Biological processes, such as responses to stress and external, abiotic, and chemical stimuli were greatly enriched among the identified DEGs, as were catalytic and oxidoreductase activities (Figure 6D). These transcriptomic changes could contribute to the high survival rate observed in transgenic *Arabidopsis* plants that had been subjected to drought stress.

Among the 55 DEGs, 17 genes that overlapped between OE-N/OE-D and WT-D/OE-D were regulated by drought stress and *ZmADF5* (Figure 6A, Table 1). Kyoto Encyclopedia of Genes and Genomes (KEGG) pathway analysis showed that these genes were enriched in various pathways, including carotenoid biosynthesis (ko00906) and circadian rhythms (ko04712). Two genes (*CYP707A3* and *PHYB*) were annotated in both pathways. *CYP707A3* is involved in carotenoid biosynthesis (ko00906), which is an important ABA signal transduction pathway. *PHYB* is involved in ko04712; it is also an ABA-dependent drought-resistant gene [34]. Six significantly altered genes were chosen for qRT-PCR analysis to confirm the DEGs identified by RNA-Seq. As shown in Appendix A, the expression levels of all the tested genes in the different materials revealed a transcription expression pattern that was similar to the RNA-Seq data.

Previous experiments have shown that ABA can induce the expression of *ZmADF5* (Figure 2C). ABA plays an important role in plant responses to drought stress as a plant hormone, and it is directly associated with both *CYP707A3* and *PHYB*, so we examined the ABA sensitivities of *ZmADF5*-overexpressing plants. In the absence of ABA, the seed germination rates of the *ZmADF5*-overexpressing lines and WT were not significantly different. With the increase in ABA concentration, the seed germination rates of WT and *ZmADF5*-overexpressing lines decreased, and *ZmADF5*-overexpressing lines displayed lower seed germination rates than WT (Appendix A). For root length measurements, the WT and *ZmADF5*-overexpressing lines had similar root length growth rates in the absence of ABA. However, the root length of *ZmADF5*-overexpressing lines was significantly longer than that of the WT under 20 μM ABA treatment (Appendix A). These experiments demonstrate that *ZmADF5*-overexpressing lines were more sensitive to ABA. Overall, these results suggest that drought tolerance in the transgenic plants might have been enhanced through an ABA-dependent pathway that led to increased expression levels of many stress-responsive genes.

## 3. Discussion

### 3.1. The Drought-Tolerant Candidate Gene ZmADF5 belongs to the Maize ADF Gene Family

Drought stress resistance in plants is a complex trait controlled by numerous QTL. Although many drought-resistance-related QTL have been identified, few genes have been cloned through QTL mapping in maize. Association analysis is widely used to identify traits and genes; several drought-related genes have been identified by association analysis, such as *ZmNAC111* [35], *ZmVPP1* [36], and *ZmTIP1* [37]. In this study, we confirmed the QTL identified by a meta-analysis and association mapping; we identified the drought-tolerance gene *ZmADF5* in maize bin 1.03, where a hotspot region for maize drought tolerance has been reported in previous studies [2,5]. *ZmADF5* belongs to the ADF gene family, which reportedly has an important role in the plant stress response. Phylogenetic analysis showed that *ZmADF5* was clustered in Group IV, where all ADF5s from different plant species were in the same branch (Appendix A). Among these genes, *AtADF5* [28] and *OsADF5* [38] conferred drought tolerance; thus, we initially presumed that *ZmADF5* might have similar drought tolerance in maize. In our study, *ZmADF5* was induced by water-deficit and ABA treatments (Figure 2B,C). The overexpression of *ZmADF5* increased drought tolerance in maize (Figure 4A,C). These results suggest that *ZmADF5* is a candidate gene for the QTL in bin 1.03 and is involved in the drought stress response.

### 3.2. ZmADF5 Contributes to the Regulation of Drought Tolerance through an ABA-Dependent Pathway Involving ROS Scavenging in Maize

Abiotic stress usually induces the accumulation of ROS in plant cells, causing damage to plants. The removal of excess ROS improves abiotic stress tolerance in plants. Antioxidants and ROS-scavenging enzymes (e.g., SOD and CAT) are essential for detoxifying ROS during normal metabolism, particularly during stress [39,40]. In our study, we found that SOD and CAT activities, as well as chlorophyll content, were significantly greater in the *ZmADF5*-OE lines than in the control plants, while the *ZmADF5*-KO lines showed contrasting results (Figure 4E–G and Figure 5E–G). Moreover, MDA is the final decomposition product of membrane lipid peroxidation caused by ROS, and its content can reflect the degree of membrane damage. Compared with the wild type, *ZmADF5*-OE plants accumulated less MDA after water-deficit treatment (Figure 4H), which indicated that the membrane damage of overexpressed *ZmADF5* plants was lower. These results demonstrate that drought stress caused less damage to the *ZmADF5*-OE lines than to Zheng58.

In our study, transgenic *Arabidopsis* overexpressing *ZmADF5* exhibited greater sensitivity to exogenous ABA than the WT plants did (Appendix A). Moreover, there is evidence that *AtADF5* [28] and *OsADF5* [38] enhance plant drought tolerance through an ABA-dependent pathway. Therefore, we presumed that *ZmADF5* may also be involved in the ABA-dependent pathway in response to drought stress. By RNA-Seq in *Arabidopsis*, among the DEGs after the water-deficit treatment, 17 (0.4%) were regulated by *ZmADF5* and the drought treatment. Among these 17 genes, 4 were involved in the ABA-dependent drought stress response. *RAP2-6* has been identified as an ABA-dependent abiotic response gene; the overexpression of *RAP2-6* enhances drought and salt stress resistance in transgenic plants [41,42]. *MSL3* is an osmotic stress-response gene; *msl2 msl3* mutants express proline and ABA metabolism genes under drought or osmotic stress [43]. *CYP707A3* is a key gene in the ABA catabolic pathway; it has an important role in the abiotic stress response, including the response to drought [44]. Moreover, the overexpression of *ZmADF5* makes transgenic *Arabidopsis* hypersensitive to ABA (Appendix A). These results show that *ZmADF5* the may be involved in ABA-dependent signaling pathway in response to drought stress.

In conclusion, our results show that *ZmADF5* is a drought stress-responsive gene in maize. *ZmADF5* can improve drought tolerance in transgenic *Arabidopsis* and maize. Under water-deficit stress, overexpression of *ZmADF5* increased SOD and CAT activities, and decreased MDA and anthocyanin accumulation. The stomatal aperture of transgenic *ZmADF5 Arabidopsis* plants decreased after water-deficit treatment, the germination rate decreased, and the root length increased after different concentrations of ABA treatment. These findings suggest that *ZmADF5* serves as a positive regulator of drought stress and has good potential to improve the genetics of drought stress tolerance in crops.

## 4. Materials and Methods

### 4.1. Association Analysis of Grain Yield in Maize under Drought Stress

In our previous study, a consensus drought-tolerance QTL was identified in maize by meta-analysis; it was named DCQ1. This QTL integrated 21 drought-tolerant QTLs with a series of backgrounds deriving from different genetic populations [2]. This consensus QTL region is located on chromosome bin 1.03; the linked marker corresponding to drought tolerance at the peak value of the confidence interval (i.e., the most likely position) was *umc1073*, which belongs to the gene *GRMZM2G077942* encoding an actin-depolymerizing factor 5 (*ZmADF5*) protein (Figure 1A). Therein, association analysis of maize grain yield (GY) under WW and WS conditions was performed using a maize-association panel that consisted of 210 inbred lines collected from China [45,46]. The plant GY was phenotyped in 2009 and 2011 in Xinjiang, China (43°54′ N, 87°28′ E), and in 2010 in Hainan, China (18°14′ N, 109°31′ E). The plants were given sufficient water during the growth period; watering was stopped before pollen dispersal for drought-stress treatment [45]. The experiments were repeated twice with both water conditions each year. The best linear, unbiased prediction of GY was evaluated for each individual in the six environments (year × location × treatment) using META-R software (http://hdl.handle.net/11529/10201) (accessed on 1 February 2018), with water management as a grouping factor.

The panel was genotyped using the Illumina MaizeSNP50 BeadChip; 41,087 single nucleotide polymorphisms with a minor allele frequency > 0.05 were included in the genome-wide association study. Mixed linear modeling with population structure (Q) and kinship (K) was conducted in TASSEL 5.0 software [47]. Principal component analysis and kinship of the panel were also calculated in TASSEL 5.0; the first four principal components were used to estimate the population structure, which explained 22.06% of the variance. Single nucleotide polymorphisms with a *p*-value < 10^−3^ were significantly associated with GY.

### 4.2. Phylogenetic Analysis of ADF Genes

In total, 78 ADF genes in maize and six other plant species (sorghum [*Sorghum bicolor*], millet [*Setaria italica*], rice [*Oryza sativa*], Arabidopsis [*Arabidopsis thaliana*], soybean [*Glycine max*], and potato [*Solanum tuberosum*]) were used for the alignment and phylogenetic analyses (Appendix A). The amino acid sequences of all ADF genes were downloaded from the National Center for Biotechnology Information (NCBI, https://www.ncbi.nlm.nih.gov/ accessed on 10 February 2019). The evolutionary analysis was conducted in MEGA7 [48] using the neighbor-joining (NJ) method [49] and 78 amino acid sequences. The optimal tree with the sum of branch length = 7.14 is shown. The percentages of replicate trees in which the associated taxa were clustered together in the bootstrap test (1000 replicates) are shown next to the branches [50]. The evolutionary distances were computed using the p-distance method [51]; the units comprised the number of amino acid differences per site. Conserved motifs in 78 ADF amino acid sequences were analyzed using Multiple Em for Motif Elicitation software (https://meme-suite.org/tools/meme accessed on 20 April 2020), and the maximum number of motifs was 20. All genome sequences were downloaded from NCBI. The conserved motif and gene structure diagrams were drawn for all ADF genes using TBtools software (https://github.com/CJ-Chen/TBtools accessed on 25 April 2020).

### 4.3. Expression Profiles of Maize ZmADF5

The gene expression data of the ADF genes in maize were downloaded from MaizeGDB (https://www.maizegdb.org/ accessed on 10 January 2019). The expression levels of seeds, roots, stems, leaves, and the flowering parts at different growth stages were analyzed using the *heatmap2* package in R software (R Foundation for Statistical Computing, Vienna, Austria). To further characterize the *ZmADF5* expression patterns among stress treatments, we conducted quantitative real-time polymerase chain reaction (qRT-PCR) analyses. Maize seeds were grown to the V2 stage in quartz sand, and then transferred to a hydroponic system in Hoagland nutrient solution. The seedlings were treated with 20% PEG6000 and 100 μM ABA for 12 h at the V3 stage. The roots, stems, and leaves of maize were sampled at 0, 1, 3, 6, and 12 h after treatment; they were compared with the corresponding controls planted in normal conditions. Tissues equally mixed from five plants were regarded as a single replicate; two replicates were collected for the treated and control materials. Total RNA was extracted from the samples using TranZol UP (TransGen Biotech, Beijing, China), in accordance with the manufacturer’s instructions. First-strand cDNA was synthesized using the FastQuant RT Kit (Tiangen). qRT-PCR was conducted using the SuperReal PreMix Plus kit (SYBR Green) (Tiangen) in IQ5 (Bio-Rad Laboratories Inc., Hercules, CA, USA). Relative expression levels were calculated using the 2^−ΔΔCt^ method [52]. The specific primers used in this study are listed in Appendix A.

### 4.4. Subcellular Localization of ZmADF5

The *ZmADF5* coding region without the terminal codon (TGA) was amplified using a pair of primers that contained *Bam*HI sites and 18 bp overlapping homologous ends of the pAN580 vector. Using the Seamless Assembly Cloning Kit (Clone Smarter), the PCR product was fused upstream of the GFP gene to generate a *ZmADF5-GFP* fusion construct driven by the CaMV35S promoter. Maize protoplasts were extracted and transformed in accordance with the method established by Yoo et al. (2007) [53]. The (35S:*ZmADF5-GFP*) vector and the control vector were transformed into maize mesophyll protoplasts using a PEG-mediated transformation method. The subcellular location of ZmADF5 was detected by fluorescence microscopy (LSM980; ZEISS, Jena, Germany) after 16 h of incubation in the dark. The specific primers used in this study are listed in Appendix A.

### 4.5. Transgenic-Positive Plant Construction and Screening

In this study, *A. thaliana* ecotype Col-3 was used as the wild type (WT). To construct the pCHF3-*ZmADF5* vector, the full-length coding sequence of the *ZmADF5* gene was amplified using KOD FX polymerase (Toyobo, Tokyo, Japan); genomic DNA from the B73 inbred line served as the template. The vector was digested with *Sma*I (New England Biolabs, Ipswich, MA, USA); the linearized vector and the *ZmADF5* DNA fragment were assembled using the Seamless Assembly Cloning Kit (Clone Smarter), in accordance with the manufacturer’s instructions. The reconstructed vector was introduced into *Agrobacterium tumefaciens* after sequencing, and then transferred to *Arabidopsis* by the floral dip method [54]. T_0_ generation seeds of transgenic plants were screened on 1/2 Murashige and Skoog (MS) medium with kanamycin. The positive plants were then transplanted into pots filled with a 2:1 mixture of vermiculite and nutritional soil. Three parallel lines of T_3_ transgenic generation plants were used for further analysis.

The *ZmADF5* coding sequence was amplified from the B73 cDNA library and inserted into the *Sma*I site of the CUB vector (this vector is controlled by the maize *UBI* promoter). The reconstructed UBI:*ZmADF5-GFP* plasmid was transformed into *A. tumefaciens*, and then introduced into the maize HiII hybrid line. Transgenic T_0_ plants were cultured in a 16 h light/8 h dark greenhouse; transgene-positive plants were detected by PCR. T_0_ transgene-positive plants were backcrossed to the Zheng58 inbred line to a purified genetic background; two independent *ZmADF5* transgenic lines were obtained for further analysis of drought resistance. CRISPR/Cas9 was used to simultaneously edit the *ZmADF5*, and then obtain knockout lines. The same guide RNA with high specificity was designed based on the *ZmADF5* coding region. The sequence was cloned into the CPB-Cas9 vector. T_0_ transgene-positive plants were identified by specific primer PCR. T_2_ seeds were obtained from T_0_ transgene-positive plants through self-crossing for further analyses. The constructed transgenic maize is listed in Appendix A.

### 4.6. Stress Treatment and Phenotyping of Transgenic Arabidopsis

The drought treatment was conducted in medium and soil. Seeds from the T_3_ generation of transgenic plants and the WT were sterilized in 10% sodium hypochlorite for 10 min, and then washed 10 times in distilled water. The seeds were planted on 1/2 MS medium plate with 1% sucrose and 0.8% agar. After 3 days of vernalization at 4 °C, the plants were grown under a long-day condition (16 h light/8 h dark) at 22 °C.

Plants for the drought-tolerance assay were transferred from 1/2 MS medium into pots filled with a 2:1 mixture of vermiculite and nutritive soil at the two-leaf stage. After 3 weeks of growth in the normal-watering condition, the plants were exposed to drought stress by the cessation of watering for 2 weeks. Then, the plants were allowed to recover via rehydration; their growth state was recorded 3 days later. The leaves of the WT and transgenic plants under drought stress and the normal-watering control were sampled. Leaves sampled from three plants were mixed as a single replicate; WT and transgenic *Arabidopsis* were subjected to five replicate water treatments. The leaves were used for stomatal aperture analysis, where leaves were floated in solutions containing 30 mM KCl and 10 mM MES-Tris, and the irradiance was set to 150 μmol·m^−2^·s^−1^ for 3 h. The width and length of the stomatal pores, as determined by using ImageJ software (http://rsbweb.nih.gov/ij accessed on 15 May 2021), were used to calculate the stomatal apertures (ratio of width to length).

### 4.7. Germination Assay and Root Growth Measurement

The sterilized *Arabidopsis* seeds were plated on 1/2 MS medium with 0 μM ABA, 0.5 μM ABA, 1.0 μM ABA, and 1.5 μM ABA. After vernalization for 3 days at 4 °C, the plates were transferred to 22 °C for culture and the germination rate of the seeds was recorded. For root length measurement, the vernalized seeds were grown on 1/2 MS medium for 4 days. Then, roots with the same length were transferred to 1/2 MS medium containing 20 μM ABA. During the growth, the root length was observed and photographed.

### 4.8. Phenotypic Analyses in Transgenic Maize

The seedlings of control and transgenic maize were germinated in Petri dishes for 3 days under dark conditions. After germination, control and transgenic maize seeds were transplanted in seedling trays filled with fertile soil and placed in a greenhouse (16 h light/8 h dark photoperiod, 25 °C). When the seedlings grew to the three-leaf stage, they were treated with drought stress by withholding watering. The plants received water after approximately 14 days, and the survival rate of the plants was determined 3 days later. Leaves of the 14-day-old maize were collected. The leaves were weighed after 0, 0.5, 1, 1.5, 2, and 4 h at room temperature, respectively. The leaves were then dried in an oven at 65 °C for 24 h; their dry weights were recorded and the water-loss rates were calculated as water loss (%) = (initial fresh weight − final fresh weight)/initial fresh weight × 100. Chlorophyll content was determined using the SPAD-502 Chlorophyll Meter. The MDA content, CAT activity, and SOD activity were determined using biochemical assay kits (Solarbio, Beijing, China). Anthocyanin content was measured using the method established by Drumm-Herrel and Mohr (1982) [55]. All samples had three biological replicates. Normally watered plants were used as the control in all experiments.

### 4.9. RNA-Seq Analysis of Transgenic Plants

Two-week-old transgenic *Arabidopsis* and WT seedlings were treated without watering for 10 days. Four groups of pooled tissues from three plants in the transgenic and WT lines were collected before and after the drought treatment. Three replicates and 12 total samples were collected for the RNA-Seq analysis. Total RNA was isolated using the RNAprep pure Plant Kit (Tiangen); the concentration and integrity of the RNA were evaluated using a NanoDrop2000 spectrophotometer (Thermo Scientific, Waltham, MA, USA) and an Agilent 2100 Bioanalyzer (Agilent Technologies, Palo Alto, CA, USA), respectively. The pooled cDNA library was sequenced using a HiSeqTM2500/4000 instrument (Illumina, San Diego, CA, USA) at Allwegene Technology (Beijing, China). In total, 53.25 Gb of raw data were obtained with a mean of 4.44 Gb data per sample. TopHat2 software was used for sequence alignment after quality control had been conducted. DEGs were identified using the DESeq method [56]. GO analysis of the DEGs was conducted using agriGO online software [57] (http://bioinfo.cau.edu.cn/agriGO/ accessed on 20 May 2021). The RNA-seq data of this study have been deposited in NCBI Sequence Read Archive (SRA, https://www.ncbi.nlm.nih.gov/sra/ accessed on 28 June 2022) with accession number PRJNA844573.

## Figures and Tables

**Figure 1 plants-13-00619-f001:**
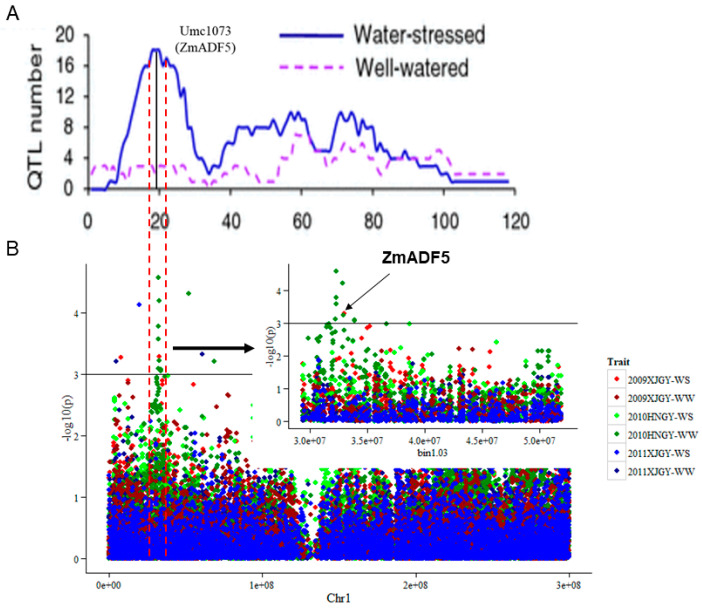
Identification of the *ZmADF5* gene through linkage mapping and association mapping meta-analysis. (**A**) Meta-analysis of constitutive QTL for drought tolerance on maize chromosome 1 (adapted from Hao et al., 2010 [2]). (**B**) Manhattan plot of the GWAS result of maize grain yield in the well-watered (WW) and water-stressed (WS) conditions.

**Figure 2 plants-13-00619-f002:**
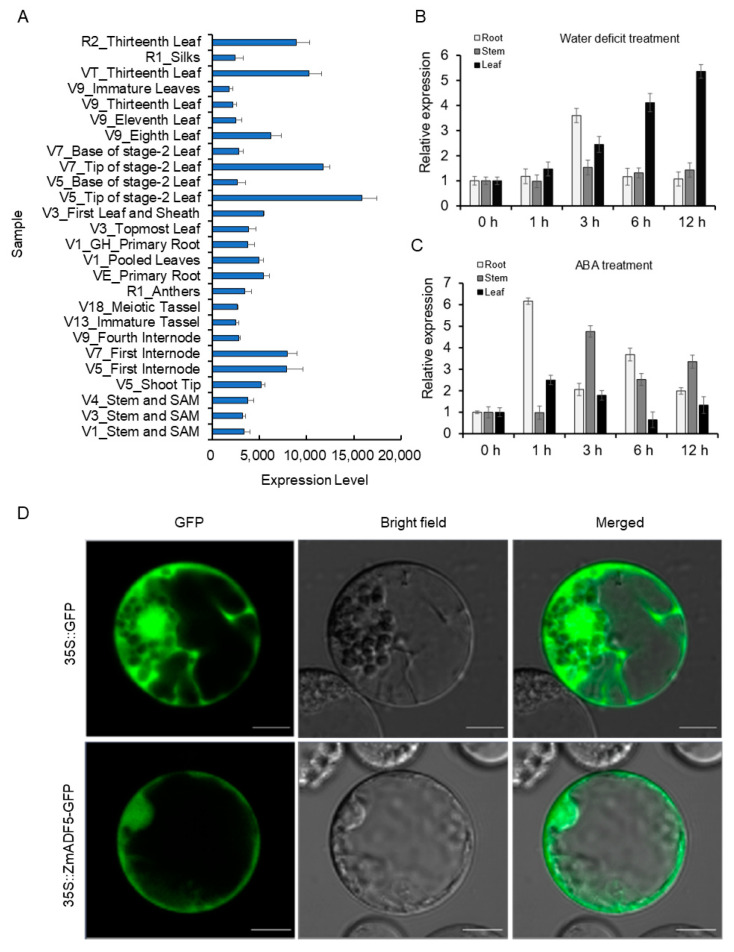
Expression patterns of *ZmADF5* in maize and subcellular localization of ZmADF5. (**A**) Expression levels of *ZmADF5* in different tissues and growth stages in the normal condition. Data were downloaded from MaizeGDB (https://www.maizegdb.org/) (accessed on 10 January 2019). (**B**) Expression pattern of *ZmADF5* at the seedling stage in the water-deficit treatment. (**C**) Expression pattern of *ZmADF5* at the seedling stage in the ABA treatment. (**D**) Localization of GFP and the ZmADF5-GFP fusion protein in maize protoplasts. Scale bar, 10 μm.

**Figure 3 plants-13-00619-f003:**
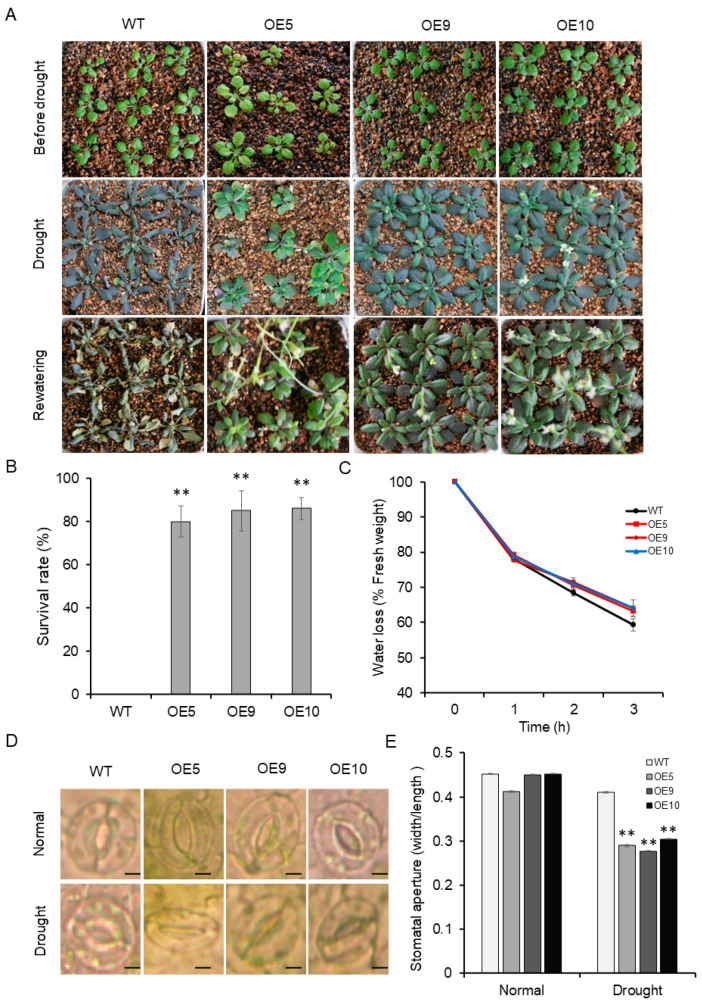
Phenotype of the 35S:*ZmADF5* transgenic *Arabidopsis*. (**A**) Drought tolerance of transgenic *Arabidopsis* plants overexpressing *ZmADF5*. (**B**) Statistical analysis of the survival rate after water-deficit stress treatment. (**C**) Water loss from detached rosettes of WT and 35S:*ZmADF5* transgenic plants. Water loss was expressed as the percentage of initial fresh weight. (**D**) Stomatal closure in WT and 35S:*ZmADF5* transgenic plants. (**E**) Values are mean ratios of width to length. Error bars represent standard errors of three independent experiments. Bars, 10 mm. Significant differences were determined by *t*-tests. ** *p* < 0.01.

**Figure 4 plants-13-00619-f004:**
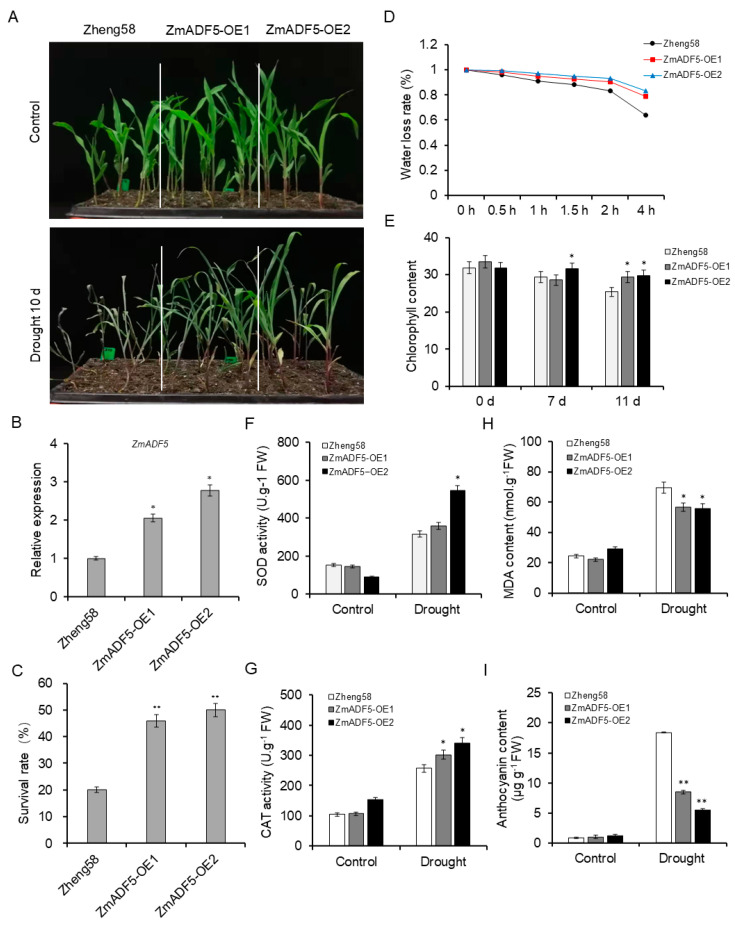
*ZmADF5* enhances drought tolerance in transgenic maize. (**A**) Drought tolerance in *ZmADF5*-OE lines. (**B**) Relative expression levels of *ZmADF5* in the transgenic maize (*ZmADF5*-OE1 and *ZmADF5*-OE2) lines under normal conditions. (**C**) Statistical analysis of survival rates after water-deficit stress treatment. (**D**) Water loss from detached rosettes of wild-type Zheng58 plants and the *ZmADF5*-OE lines. Water loss was expressed as the percentage of initial fresh weight. (**E**–**I**) Comparison of chlorophyll content (**E**), SOD activity (**F**), CAT activity (**G**), MDA content (**H**), and anthocyanin content (**I**) between leaves from wild-type Zheng58 plants and *ZmADF5*-OE lines under water-deficit stress conditions. Significant differences were determined by *t*-tests. * *p* < 0.05, ** *p* < 0.01.

**Figure 5 plants-13-00619-f005:**
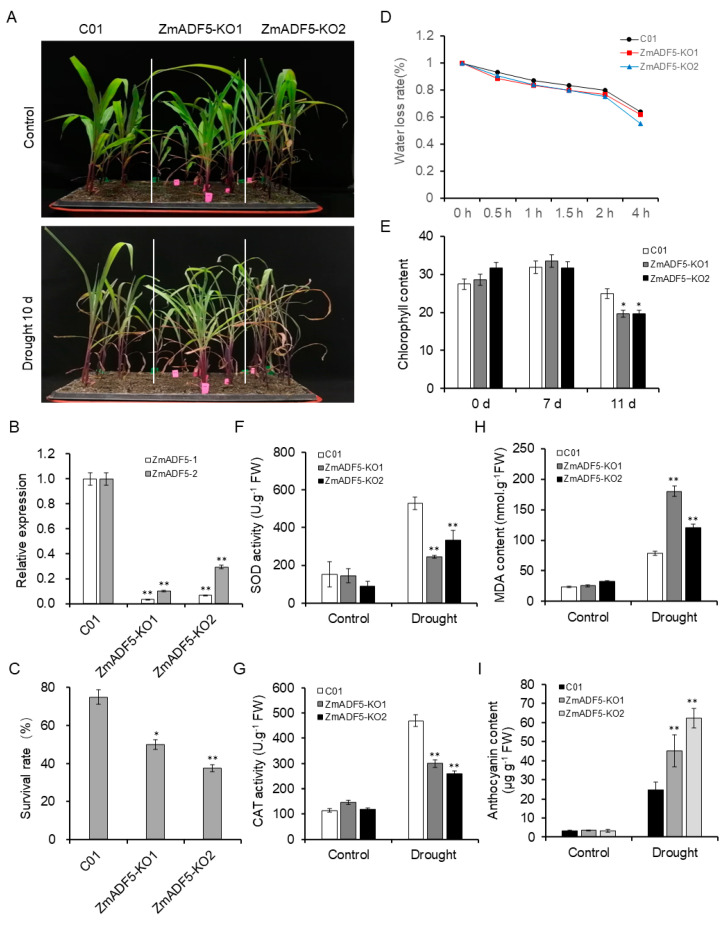
Knocking out *ZmADF5* reduces drought tolerance in transgenic maize. (**A**) Drought tolerance in *ZmADF5*-KO lines. (**B**) Relative expression levels of *ZmADF5* in *ZmADF5* knockout (*ZmADF5*-KO1 and *ZmADF5*-KO2) lines under normal conditions. (**C**) Statistical analysis of the survival rate after water-deficit stress treatment. (**D**) Water loss from detached rosettes of wild-type CO1 plants and the *ZmADF5*-KO lines. Water loss was expressed as the percentage of initial fresh weight. (**E**–**I**) Comparison of chlorophyll content (**E**), SOD activity (**F**), CAT activity (**G**), MDA content (**H**), and anthocyanin content (**I**) between leaves from wild-type CO1 plants and the *ZmADF5*-KO lines under water-deficit stress conditions. Significant differences were determined by *t*-tests. * *p* < 0.05, ** *p* < 0.01.

**Figure 6 plants-13-00619-f006:**
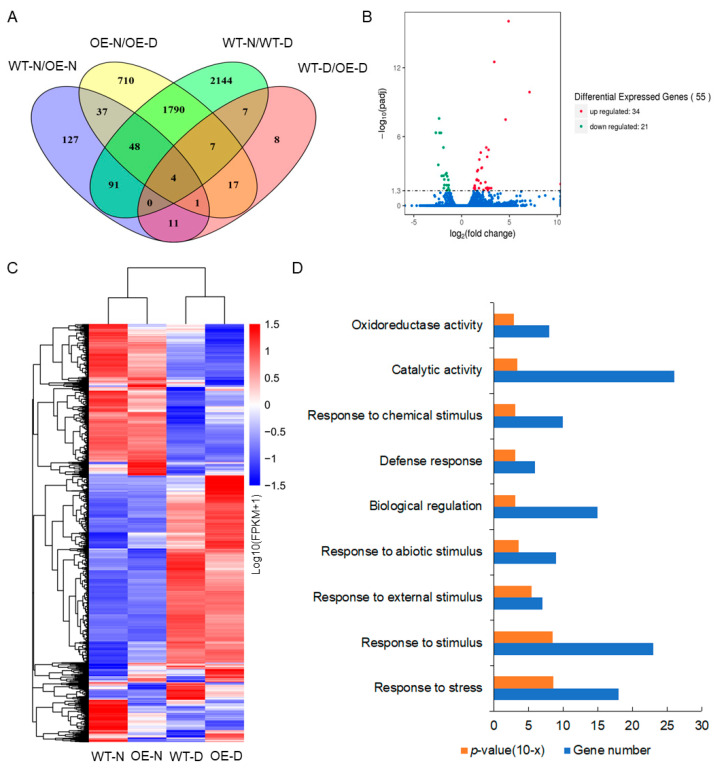
Transcriptomic analysis of 35S:*ZmADF5* transgenic *Arabidopsis* under normal and water-deficit conditions. (**A**) Venn diagrams of differentially expressed genes in OE plants relative to WT plants under normal-watering and water-deficit conditions, using a significant cutoff of *p*-value < 0.05. (**B**) Volcano plot of differentially expressed gene distributions in *ZmADF5*-overexpressing and wild-type *Arabidopsis* plants under water-deficit stress. The red and green dots indicate significantly differentially up-regulated and down-regulated genes, respectively. The blue dots represent non differentially expressed genes. (**C**) Heatmap and cluster analysis of differentially expressed genes. (**D**) GO annotation of differentially expressed genes in *ZmADF5*-overexpressing and wild-type *Arabidopsis* plants under water-deficit stress.

**Table 1 plants-13-00619-t001:** Differentially expressed genes are regulated by *ZmADF5* overexpression and drought stress.

	Gene_ID	Change ^1^	Gene Name	Gene Description
Upregulated	AT1G43160	4.91	*RAP2-6*	Ethylene-responsive transcription factor RAP2-6
	AT1G51780	7.10	*ILL5*	IAA-amino acid hydrolase ILR1-like 5
	AT1G58200	1.30	*MSL3*	MSCS-like 3
	AT2G36590	2.49	*PROT3*	Proline transporter 3
	AT2G38240	4.60	*ANS*	2-oxoglutarate (2OG) and Fe (II)-dependent oxygenase superfamily protein
	AT3G57460	3.42		Catalytic/metal ion binding/metalloendopeptidase/zinc ion binding protein
	AT4G23060	1.56	*IQD22*	IQ-domain 22
	AT5G27060	2.75	*AtRLP53*	Receptor-like protein 53
	AT5G62360	1.86		Plant invertase/pectin methylesterase inhibitor superfamily protein
	AT1G30320	1.81		Remorin family protein
	AT1G52720	1.32		Hypothetical protein
	AT2G18790	1.37	*PHYB*	Phytochrome B
	AT3G05640	1.63		Protein phosphatase 2C family protein
	AT4G35110	1.39		Phospholipase-like protein (PEARLI 4) family protein
Downregulated	AT5G45340	−2.36	*CYP707A3*	Cytochrome p450 family, family 707, subfamily A, polypeptide 3
	AT5G61600	−2.16	*ERF104*	Ethylene response factor 104
	AT1G07135	−1.36		Glycine-rich protein

^1^. |log2 (FoldChange)| ≥ 1 and q-value ≤ 0.05.

## Data Availability

Data are contained within the article and Appendix A.

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
