# Peer review of "ZmADF5, a Maize Actin-Depolymerizing Factor Conferring Enhanced Drought Tolerance in Maize"

_plants, 2024, doi:10.3390/plants13050619_

Round 1
Reviewer 1 Report
Comments and Suggestions for Authors
1. At line no. 18 “significant signals related it were also identified” rephrase this sentence.
2. At line no. 47 Check the name of trait “ear location” OR ear height (ref. no. 10).
3. In Figures 2B and 2C caption also mentions “drought treatment” and “ABA treatment”.
4. At line no. 185 correct “ZmADF5-KO and ZmADF5-OE lines” to “ZmADF5-OE and ZmADF5-KO lines.
5. Expand the parameters such as CAT and SOD activities, MDA contents, and WT plants first time, and then use abbreviations subsequently throughout the manuscript.
6. Provide supplementary figures and table captions in the Supplementary Materials section of the manuscript.
Comments on the Quality of English LanguageThe English Language is fine. Some minor corrections are required.
Author Response
(1) Comment 1: At line no. 18 “significant signals related it were also identified” rephrase this sentence.
Response to comment 1: Thanks for your suggestion. We have rephrased this sentence as “and the significantly associated signals were detected through genome wide association analysis.” (Lines 18 to 19)
(2) Comment 2: At line no. 47 Check the name of trait “ear location” OR ear height (ref. no. 10).
Response to comment 2: Thanks for your comments. We have checked the name of this trait "ear location" and we found it is called ear height in most of papers, which has been changed in the article. (Lines 48). As in reference No. 10, this trait was mentioned by the name “ear height”.
(3) Comment 3: In Figures 2B and 2C caption also mentions “drought treatment” and “ABA treatment”.
Response to comment 3: Thanks for your comments. We have revised and added "Drought treatment" and "ABA treatment" to the caption in Figures 2B and 2C.
Fig. 2 Expression patterns of ZmADF5 in maize and subcellular localization of ZmADF5.
(4)Comment 4: At line no. 185 correct “ZmADF5-KO and ZmADF5-OE lines” to “ZmADF5-OE and ZmADF5-KO lines.
Response to comment 4: Thanks for your comments. Based on your suggestion, we have corrected “ZmADF5-KO and ZmADF5-OE lines” to “ZmADF5-OE and ZmADF5-KO lines. (Lines 186)
(5)Comment 5: Expand the parameters such as CAT and SOD activities, MDA contents, and WT plants first time, and then use abbreviations subsequently throughout the manuscript.
Response to comment 5: Agree. Taken your suggestion. We have expanded the parameters such as CAT and SOD activities, MDA contents, and WT plants when they appeared in the manuscript for the first time as wild type (WT) plants, the malondialdehyde (MDA) content, the catalase (CAT) activity, and the superoxide dismutase (SOD) activity, and then use their abbreviations subsequently throughout the manuscript. (Lines 160, 197, 198)
(6)Comment 6: Provide supplementary figures and table captions in the Supplementary Materials section of the manuscript.
Response to comment 6: Agree. Taken your suggestion. We have further checked and provided supplementary figures and table captions in the Supplementary Materials section of the manuscript. As the following:
“Supplementary Materials: The following supporting information can be downloaded at: www.mdpi.com/xxx/s1.
Figure S1: Phylogenetic analysis, conserved motifs, and gene structural analysis of all ADF genes.
Figure S2: Amino acid sequence alignment of two ZmADF5 genes.
Figure S3: Target site design of ZmADF5 gene knockout and PCR identification of the knockout lines.
Figure S4: Expression levels of differentially expressed genes in different plants under normal and drought conditions.
Figure S5: ABA sensitivity of 35S:ZmADF5 transgenic Arabidopsis plants during germination and post-germination growth.
Table S1: Information regarding all 78 ADF genes in maize and six other plant species.
Table S2: Primers used in this study.
Table S3: Transgenic maize used in this study. (line 504-512)
- Response to Comments on the Quality of English Language
Point 1: The English Language is fine. Some minor corrections are required.
Response to Point1: Thanks for your comments. The questions and suggestions raised have been revised and the article has been checked again.
- Additional clarifications
Additional clarifications:
Lines 201,207,316,472,473. We modify “wild type” to “control”, make it easier to distinguish the wild type contrast between Arabidopsis and maize.
Line 273, We deleted the “N” and “D”, We revised this sentence as “in OE plants relative to WT plants under normal and drought conditions”
Lines 211, 235, 239,246 references to reactive oxygen species (ROS), differentially expressed genes (DEGs), Gene Ontology (GO), and Kyoto Encyclopedia of Genes and Genomes (KEGG) are explained when they first appear, and lines 311, 499 use abbreviations
We checked this manuscript and removed reference 38, which was not necessary for our study. Based on this, we readjusted the position of the references.

Reviewer 2 Report
Comments and Suggestions for Authors
It's an interesting paper and the results are relevant for work to improve crop productivity under climate change.
I only have a few comments and suggestions to clarify what has been done and to make the paper a little easier to read.
The results are good and well presented. However, why is the expression of ZmADF5 (presumably detecting both 5-1 and 5-2?) detailed in Figures 2 and 4 but in Figure 5 there is data for both ZmADF5-1 and ZmADF5-2. Can more data be added to Figures 2 and 4?
Comments on the Quality of English Language
The English is good but it could do with some editing, I have made some suggestions below though there are other minor ones that a proof-reading will pick up.
Spelling - line 15 irrigation; line 470 seedlings;
English: In the abstract I would define again what the ADF gene is.
Also add "when" - Drought stress is seriously affecting the growth and production of crops, especially when the agricultural irrigation still remains quantitatively restricted in some arid and semi-arid areas.”
Line 32/33 - clarify the sentence "As the reduced variation in grain yield (GY) response to stress, there will be a long limited selection of superior genotypes to increase GY under drought-stressed conditions."
Line 48 - "These genes have been functionally analyzed and further exploited later afterwards." What do you mean by later afterwards? Have researchers used these in breeding programs or other studies?
Line 187 - The ZmADF5-OE lines showed better than did the control plants (Fig. 4A); however, the ZmADF5-KO plants showed the ..." Showed better what?
Line 233 – define N and D in text not just in the Figure legend eg normal watering (N) and drought (D) treatments
Author Response
(1) Comment 1: It's an interesting paper and the results are relevant for work to improve crop productivity under climate change. I only have a few comments and suggestions to clarify what has been done and to make the paper a little easier to read. The results are good and well presented. However, why is the expression of ZmADF5 (presumably detecting both 5-1 and 5-2?) detailed in Figures 2 and 4 but in Figure 5 there is data for both ZmADF5-1 and ZmADF5-2. Can more data be added to Figures 2 and 4?
Response to comment 1: Thanks for your suggestions. In the result, a significant signal only in ZmADF5-1 gene associated with drought tolerant-related traits was identified by genome-wide association analysis. An interesting result was found that ZmADF5-2 gene has a highly similar sequence with ZmADF5-1 gene of 80.81%, however ZmADF5-1 gene located in the confident interval of a drought tolerant QTL, while ZmADF5-2 was not. So the function of ZmADF5-1 was further studied and most of the work focused on it. Moreover, Huang et al. (2020) has found that the expression level of the gene named ZmADF8 (ZmADF5-2) was significantly up-regulated under drought and ABA treatments. Therefore, the relevant data could not be presented in Figure 2 and 4. However, in subsequent experiments, the function of both genes will be studied by constructing more transgenic plants to test their difference on drought tolerance.
Reference: Huang, J., Sun, W., Ren, J.X., Yang, R.C., Fan, J.S., Li, Y.F. (2020) Genome-Wide Identification and Characterization of Actin-Depolymerizing Factor (ADF) Family Genes and Expression Analysis of Responses to Various Stresses in Zea Mays L. Int. J. Mol. Sci. 21, 1751; doi:10.3390/ijms21051751.
(2) Comment 2: The English is good but it could do with some editing, I have made some suggestions below though there are other minor ones that a proof-reading will pick up.
Spelling - line 15 irrigation; line 470 seedlings;
Response to comment 2: Thanks very much for your suggestions. We have revised the spelling of line 15 and line 470. “irrigaiton to irrigation” (line 15) “seedings to seedlings” (line 472,488). And we have further checked the full text and done some revisions.
(3) Comment 3: English: In the abstract I would define again what the ADF gene is.
Response to comment 3: Thanks for your suggestions. ZmADF5 is a member of the actin depolymerization factor family. We have defined what the ADF gene is in the abstract again, described in detail in introduction. Therefore, we revised “Here, we found that a new member of actin-depolymerizing factor (ADF) family, ZmADF5 gene was tightly linked with…”(Lines 17).
(4) Comment 4: Also add "when" - Drought stress is seriously affecting the growth and production of crops, especially when the agricultural irrigation still remains quantitatively restricted in some arid and semi-arid areas.”
Response to comment 4: Thanks for your suggestions. We have added "when" in the manuscript, so the sentence was revised as “Drought stress is seriously affecting the growth and production of crops, especially when the agricultural irrigation still remains quantitatively restricted in some arid and semi-arid areas.” (Lines 14).
(5) Comments 5: Line 32/33 - clarify the sentence "As the reduced variation in grain yield (GY) response to stress, there will be a long limited selection of superior genotypes to increase GY under drought-stressed conditions."
Response to comment 5: We apologize that our expression caused you trouble, and we explain it in detail here. We revised the sentence "As the reduced variation in grain yield (GY) response to stress, there will be a long limited selection of superior genotypes to increase GY under drought-stressed conditions." to “As the reduced variation in grain yield (GY) response to stress, the selection of superior genotypes for GY under drought-stressed conditions usually has limited and inconsistent progress”, Which it showed be clearer than before(Line 33to 35).
(6) Comment 6: Line 48 - "These genes have been functionally analyzed and further exploited later afterwards." What do you mean by later afterwards? Have researchers used these in breeding programs or other studies?
Response to comment 6: We are very sorry for the misrepresentation here Line 48. Most of the QTLs or candidate genes associated with traits mentioned in references. No. 7,10,11,12,13 have been identified, and subsequent gene cloning and functional verification have not been carried out. But these findings could provide important information for future molecular mechanisms. Therefore, we have deleted this sentence “"These genes have been functionally analyzed and further exploited later afterwards." and amended it to "However, few of them have been cloned or functionally analyzed." (Lines 49).
(7) Comments 7: Line 187 - The ZmADF5-OE lines showed better than did the control plants (Fig. 4A); however, the ZmADF5-KO plants showed the ..." Showed better what?
Response to comment 7: We apologize that our expression caused you trouble, and we explain it in detail here. What we want to show is that after drought treatment, compared with wild type, the leaves of ZmADF5-OE lines are normal, still maintain green, and grow well. We modify this sentence as” The ZmADF5-OE lines showed normal with leaves that remained green and grew better than did the control plants (Fig. 4A); However, the ZmADF5-KO plants showed the ...". (line 188 to 189)
(8) Comments 8: Line 233 – define N and D in text not just in the Figure legend eg normal watering (N) and drought (D) treatments
Response to comment 8: Thanks for your suggestions. We have defined N and D in text. We modified this sentence “ZmADF5-overexpressing (OE10) transgenic line and WT plants of Arabidopsis under normal watering and drought treatments” as “ZmADF5-overexpressing (OE10) transgenic line and WT plants of Arabidopsis under normal watering and drought treatments (WT-N and OE-N under normal conditions; WT-D and OE-D under drought conditions)” (line 234 to 235).
- Response to Comments on the Quality of English Language
Point 1: The English Language is fine. Some minor corrections are required.
Response to Point1: Thanks for your comments. The questions and suggestions raised have been revised and the article has been checked again.
- Additional clarifications
Additional clarifications:
Lines 201,207,316,472,473. We modify “wild type” to “control”, make it easier to distinguish the wild type contrast between Arabidopsis and maize.
Line 273, We deleted the “N” and “D”, We revised this sentence as “in OE plants relative to WT plants under normal and drought conditions”
Lines 211, 235, 239,246 references to reactive oxygen species (ROS), differentially expressed genes (DEGs), Gene Ontology (GO), and Kyoto Encyclopedia of Genes and Genomes (KEGG) are explained when they first appear, and lines 311, 499 use abbreviations
We checked this manuscript and removed reference 38, which was not necessary for our study. Based on this, we readjusted the position of the references.
